# Drivers of Three Most Charismatic Mammalian Species Distribution across a Multiple-Use Tropical Forest Landscape of Sumatra, Indonesia

**DOI:** 10.3390/ani12192722

**Published:** 2022-10-10

**Authors:** Dede Aulia Rahman, Yanto Santosa, Intan Purnamasari, Aryo Adhi Condro

**Affiliations:** 1Department of Forest Resources Conservation and Ecotourism, Faculty of Forestry & Environment, Kampus IPB Dramaga, IPB University, Bogor 16680, Indonesia; 2Primate Research Center, Institute of Research and Community Service, Kampus IPB Lodaya, IPB University, Bogor 16151, Indonesia

**Keywords:** conservation areas, *Elephas maximus sumatranus*, forest disturbance, land use, multiple-use landscape, *Pongo abelii*, *Panthera tigris sumatrae*, tropical areas

## Abstract

**Simple Summary:**

Charismatic Sumatran mammals (i.e., the elephant, orangutan, and tiger) play a pivotal role in maintaining ecosystem balance. Nevertheless, these species have encountered multifaceted threats due to habitat disturbances, leading to their potential extinction. Thus, understanding current species ranges, together with investigating the most essential factors for the species, are crucial for developing conservation strategies. We predicted the potential range of three charismatic mammals in Sumatra Island using anthropogenic, biophysical, topographic, and climatic parameters based on ensemble machine learning algorithms. This study also comprehensively describes how land ownerships can shape the existence of these three species. Finally, we provide recommendations based on our findings for species conservation planning and management options.

**Abstract:**

Tropical Rainforest Heritage sites of Sumatra are some of the most irreplaceable landscapes in the world for biodiversity conservation. These landscapes harbor many endangered Asiatic mammals all suffering multifaceted threats due to anthropogenic activities. Three charismatic mammals in Sumatra: *Elephas maximus sumatranus*, *Pongo abelii*, and *Panthera tigris sumatrae* are protected and listed as Critically Endangered (CR) within the IUCN Red List. Nevertheless, their current geographic distribution remains unclear, and the impact of environmental factors on these species are mostly unknown. This study predicts the potential range of those species on the island of Sumatra using anthropogenic, biophysical, topographic, and climatic parameters based on the ensemble machine learning algorithms. We also investigated the effects of habitat loss from current land use, ecosystem availability, and importance of Indonesian protected areas. Our predictive model had relatively excellent performance (Sørensen: 0.81–0.94) and can enhance knowledge on the current species distributions. The most critical environmental predictors for the distribution of the three species are conservation status and temperature seasonality. This study revealed that more than half of the species distributions occurred in non-protected areas, with proportional coverage being 83%, 72%, and 54% for *E.m. sumatranus*, *P. abelii*, and *P.t. sumatrae*, respectively. Our study further provides reliable information on places where conservation efforts must be prioritized, both inside and outside of the protected area networks, to safeguard the ongoing survival of these Indonesian large charismatic mammals.

## 1. Introduction

Tropical forests are home to two thirds of the world’s biodiversity [1]. Despite representing the largest reservoir of biodiversity with a rich diversity of flora (>200 plant species/hectare) [2,3] and fauna (>50% of the world’s animal species) [4] and its various functionalities, most tropical rainforests have suffered degradation and deforestation due to human intervention over time, including those within Indonesia. As intact forest declines, species are forced to adapt to more degraded habitats and mosaics of anthropogenic land-use types. Understanding how species respond to human-modified forests provides critical information regarding land-use decisions and species-specific management strategies for conservation. The tropical rainforest landscape of Sumatra harbors 13 orders of the mammalian class: Proboscidea, Sirenia, Eulipotyphla, Chiroptera, Pholidota, Carnivora, Perissodactyla, Artiodactyla, Lagomorpha, Rodentia, Scandentia, Dermoptera, and Primates. Of those 13 orders, there are three of many highly charismatic mammals which are threatened with extinction: Sumatran elephant (*Elephas maximus sumatranus*), Sumatran orangutan (*Pongo abelii*), and Sumatran tiger (*Panthera tigris sumatrae*). Currently, all three species are categorized as Critically Endangered on the IUCN Red List (*E.m. sumatranus* [5]; *P. abelii* [6]; and *P.t. sumatrae* [7]) and listed as protected species under Indonesian regulations [8].

Sumatran wildlife has encountered increased habitat disturbances over the recent decades and the loss of habitat for those three charismatic mammals is a major threat to their survival. The development of forest resources, which have helped Indonesia to achieve economic growth, has resulted in forest loss and decline over the last 25 years. Based on data from the 2015 Directorate General of Planology, Ministry of Environment and Forestry, between 2013–2014 Sumatra experienced one of the highest annual rates of deforestation compared to other large islands in Indonesia. While there have been important studies focused on the population and behavioral ecology of *E.m. sumatranus* (e.g., [9,10]), *P. abelii* (e.g., [11,12,13]), and *P.t. sumatrae* (e.g., [14,15,16]), there have been few studies of their ecological distribution or status and the factors driving their distribution [17,18,19,20,21,22]. With the increasing threats to Sumatran wildlife due to habitat loss and degradation, it is vital to assess their distribution and habitat use in order to prioritize protection of critical areas. Overall, there is a lack of information even in those areas reported on and this often limits the potential to infer habitat preference and distribution, much less prescribe conservation measures, for these rare and elusive species.

The use of presence-only data has become important information in wildlife research, particularly for mapping distribution of cryptic or rare charismatic species. Recently, data from various surveys have been used in ecological niche models in order to predict mammalian species distribution [23,24,25,26,27,28,29,30]. Such models are important for various applications [31,32]. Nevertheless, there is still lack of information regarding the species range with limited observational data related to these rare species. Therefore, predicting the species distribution using small amount of information is crucial. Previous studies showed that various machine learning algorithms, such as Boosted Regression Tree, Maximum Entropy, Random Forest, Support Vector Machine, and Ensemble from various algorithms had relatively good performance in predicting species distribution for the specific species in certain landscapes [33,34,35,36,37,38,39]. The use of models that utilize only presence data has been debated, but recent studies have shown that in some cases, presence-only modeling techniques may even have better predictive accuracy than more traditional presence–absence methods [40,41].

We used five algorithms to predict the species ranges of three charismatic mammalian species (*E.m. sumatranus*, *P. abelii*, and *P.t. sumatrae*) in several conservation areas in Sumatra. We then used a best model to depict the most crucial areas inside or outside of the protected areas for those charismatic species to improve conservation management and planning. Moreover, we tested the hypotheses that (i) all three charismatic species are strongly attributed to the forests as their habitat and (ii) undisturbed protected areas are essential for their conservation. We also addressed the specific question of how the distribution of these three species in Sumatra responds to the variation in multiple-use landscapes. The information gathered from this study will be a basis for future research on the island and for developing a plan for island-wide conservation of those species.

## 2. Materials and Methods

### 2.1. Study Areas

In this study, we analyzed all extents of Sumatran Island, both in protected and non-protected areas, which are likely to support the focal species, from sea level peat swamps to forests around the volcanic peak of Mount Kerinci, the highest point on Sumatra (3805 m asl). Sumatra’s network of protected areas cover around 23% of the total area (~110,000 km^2^), of which 42% are conservation areas and 58% are watershed protection reserves. There are 150 conservation areas established to preserve biodiversity on the island with 13 national parks, 48 nature reserves, and 25 wildlife sanctuaries, including three UNESCO-listed national parks, namely Gunung Leuser, Kerinci Seblat, and Bukit Barisan Selatan. However, conservation areas in Sumatra have been subjected to significant deforestation since their establishment and most of these areas are located and bordered by agricultural fields, forest concessions, oil palm plantations, and densely populated villages. Sumatra Island is one of the islands in Indonesia with among the highest deforestation rate in the world as a result of a combination of human anthropogenic activities in the form of (1) forest conversion for industrial plantations, (2) semi-forest fires, (3) road construction, and (4) small-scale forest clearing. Since 1985, Sumatra has lost 12.5 million ha of natural forest [42] with an annual conversion rate of ~500,000 hectares (2.56% yr^−1^). Most forest loss (>80%) occurred in lowland areas with convenient access, where the forests contained the most diverse ecosystems with high carbon stock. 

### 2.2. Methods

#### 2.2.1. Collection of Records

Species occurrence for *E.m. sumatranus*, *P. abelii*, and *P.t. sumatrae* were collected from the Ministry of Environment and Forestry (MoEF), Global Biodiversity Information Facility [43], IUCN Global Assessment species ranges [44], fieldwork and reports from monitoring projects in Indonesia. We compiled 8866 records of Sumatran elephant, 670 records of Sumatran orangutan, and 1199 records of Sumatran tiger. To reduce sampling bias effects and spatial autocorrelation, we conducted a thinning process following [45] towards our occurrences data. We subsequently used 1222 records of Sumatran elephant, 493 records of Sumatran orangutan, and 528 records of Sumatran tiger. We generated pseudo-absence data (*n* = 10,000) in equal prevalence with random selection of geographical and environmental constraints as the absence data were not available [46].

#### 2.2.2. Environmental Descriptions

This study used 20 environmental variables to capture geographic ranges of Sumatran elephant, orangutan, and tiger and the response of those three species to land dynamics and policy interventions. We divided the environmental layers into four different groups, i.e., physical, biotic, climatic, and anthropogenic (Appendix A). We used elevation, slope, aspect, eastness, and northness data as topographic variables, retrieved from the Shuttle Radar Topographic Mission (SRTM) [47]. In the biophysical group, we used several spectral indices that were derived from optical satellite imageries (Landsat 8), i.e., EVI, NDVI, NDWI, SAVI, and IBI, and protected areas. We also used binary information of land cover, access to forest plantations, logging concessions, oil palm, and social forestry as the anthropogenic group. The Indonesian maps of protected areas and land cover were retrieved from the MoEF, Climate Change Initiative–European Space Agency (CCI-ESA) [48], and World Resources Institute (WRI). In addition, we considered average and variation of temperature and precipitation, respectively, as the climatic group retrieved from WorldClim v2.0 [49] (See Appendix A for further detail of variable descriptions).

We resampled all environmental covariates to a spatial resolution of 30 arc-second (~1 km). We conducted variable selection based on Pearson correlation (|r| > 0.7) following [50] to overcome collinearity issues that can affect the model output [51]. Afterwards, we eliminated five variables from the analysis due to multicollinearity issues (i.e., elevation, aspect, EVI, NDVI, and IBI).

#### 2.2.3. Model Calibration and Evaluation

To predict species ranges for Sumatran elephant, orangutan, and tiger, we performed species distribution modelling based on various machine learning algorithms, i.e., Boosted Regression Trees (BRT), Maximum Entropy (MXD), Random Forest (RDF), and Support Vector Machine (SVM) [33,36,38,52] using ENMTML package [53]. This study performed an Ensemble (ENS) model based on the first component of principal components analysis, considered by Sørensen value [54].

This study evaluated the model using discrimination metrics: area under the curve (AUC), Kappa, true skill statistic (TSS), Jaccard, and Sørensen [55,56]. Ref. [56] revealed that similarity indices can better capture the species distribution model performance, due to the prevalence issues, than commonly used metrics (e.g., AUC or TSS). However, we still provided AUC, TSS, and inter-rater agreement metrics, since these metrics are still adopted by most researchers. We also presented Jaccard similarity index for comparation with Sorensen similarity index. To convert the probability of occurrence into binary maps for model evaluation, we used maximized Sørensen metric [56]. Moreover, we performed spatially restricted species distribution models based on a posteriori method (i.e., buffered minimum convex polygon of the occurrence data) [57] to deal with overprediction due to dispersal movement exclusion [58].

#### 2.2.4. Conservation and Threat Issues

First, we confirmed the distribution of the three charismatic mammal species based on various reports. Then we included all protected areas, national parks, nature reserves, wildlife sanctuaries, hunting parks, nature recreational parks, and pristine reserves in our analytical method. We used the Update tool within ArcGIS 10.5 to generate a single polygon by removing all the stand-alone protected areas and merging them with the associated suitable patch. Then, we examined the suitable patches identified as an indicator of the important ecological area within each protected area to support our three focal species. The historical map of protected areas in Sumatra was developed based on historical information on protected areas in Indonesia [59,60]. The extent to which our species’ range overlapped with industrial oil palm plantations (IOPP), logging concession (LC), forest plantation concession (FPC), and social forestry concession (SFC) was assessed with the same approach as in protected areas. Following that, we examined how these plantations and/or concession areas intersected with three charismatic mammalian species in suitable landscapes in Sumatra.

In addition, we also conducted climatic exposure calculations in protected and non-protected areas to investigate the influence of climate on the three charismatic mammalian species. Current and future climatic condition were measured based on a global climate model (GCM) using the standardized Euclidean distance based on temperature and precipitation, following [61].

## 3. Results

### 3.1. Models Performance Evaluation

Through the cross-validation evaluation of the tested models, AUC, Kappa, TSS, Jaccard, and Sørensen values were obtained for the testing of each model (Figure 1). This study found that the ensemble model from four machine learning algorithms outperformed the single algorithm approaches, with the average (standard error) of Sørensen index being 0.81 (0.01), 0.94 (0.01), and 0.84 (0.03) for Sumatran elephant, orangutan, and tiger, respectively. All models performed well (AUC > 0.80, Kappa > 0.46, TSS > 0.46, Jaccard > 0.60, and Sørensen > 0.75) (Appendix A). For the testing data, the five models’ AUC values varied from 0.80 (SVM) to 0.88 (RDF and ENS), Kappa values varied from 0.46 (SVM) to 0.86 (RDF and ENS), TSS values from 0.46 (SVM) to 0.86 (RDF and ENS), Jaccard values varied from 0.60 (SVM and MXD) to 0.88 (ENS), and Sørensen values ranged from 0.75 (SVM and MXD) to 0.93 (ENS and RDF). Overall, the discrimination metrices indicated that the ensemble model (ENS) from four different algorithms (i.e., MXD, RDF, BRT, and SVM) provided the best predictive performance, while the Support Vector Machine (SVM) was the weakest, thus, we selected the ENS model to establish charismatic mammalian distribution patterns.

### 3.2. Environmental Variables and Habitat Suitability

The most important variable for most of the species was the anthropogenic aspect with an overall score of more than 11%. The climatic parameter, in this context the temperature, was also important in predicting distribution of the three species, with the variable importance overall score of 6% (Table 1). The response curves indicated that all three species were more likely to occur outside of protected areas with the temperature seasonality ranging from 20 °C to 35 °C (Appendix A). This study showed that forests, agriculture, and shrubs were the main habitat types either inside or outside the protected areas across Sumatra (Appendix A). Further detail on the variable importance for each species can be seen in Appendix A.

This study found that climate exposure outside protected areas (PAs) was significantly greater than inside PAs (*p*-value < 0.05), with the average exposure (SED) being 3.94 and 5.02 for PAs and non-PAs, respectively (Figure 2). We indicate that species outside PAs would likely suffer more intense climate exposure than species within the PAs. Furthermore, we found that the nature recreation parks suffered the highest climatic expo-sure from the other conservation functions with an average SED of about 6.65 ± 3.44. On the other hand, nature reserves had the lowest climatic exposure within the protected areas, with an average SED of about 3.32 ± 3.38 (Appendix A).

### 3.3. Suitable Landscape, Protected Areas, and Concessions

This study revealed that populations of the three species occurred in some highly fragmented areas, which are more isolated. *E.m. sumatranus*, *P. abelii*, and *P.t. sumatrae* mostly occurred in the eastern, western, and northern part of Sumatra, respectively (Figure 3A: *E.m. sumatranus*, Figure 3B: *P. abelii*, Figure 3C: *P.t. sumatrae*). The area of suitable landscape for *E.m. sumatranus*, *P. abelii*, and *P.t. sumatrae* were 5,775,350.40 ha, 2,642,585.69 ha, and 8,005,540.59 ha, respectively, and more than half of the geographical ranges for the three species were found outside protected areas, with the proportion of distributions being 83%, 72%, and 54% for *E.m. sumatranus*, *P. abelii*, and *P.t. sumatrae*, respectively. Only 17% (*E.m. sumatranus*), 28% (*P. abelii*), and 46% (*P.t. sumatrae*), of the species ranges occurred within protected areas and 27, 9, and 71 different protected areas functioned as suitable habitats for *E.m. sumatranus*, *P. abelii*, and *P.t. sumatrae*, respectively (Appendix A). Outside of protected areas, *E.m. sumatranus* distribution overlapped with FPC (25.87%), LC (2.97%), IOPP (3.37%), and SFC (3,10%). *P. abelii* distribution overlapped with FPC (3.43%), LC (2.54^−3^%), IOPP (0.81%), and SFC (2.48%), and *P.t. sumatrae* distribution overlapped with FPC (4.31%), LC (3.94%), IOPP (0.48%), and SFC (2.24%) (Appendix A). 

## 4. Discussion

The distribution of *E.m. sumatranus*, *P. abelii*, and *P.t. sumatrae* is described in the Indonesian Elephant Conservation Strategy and Action Plan 2007–2017 [62], Indonesian Orangutan Conservation Strategy and Action Plan 2007–2017 [63], and Indonesian Tiger Conservation Strategy and Action Plan 2007–2017 [64]. This study provides a comprehensive, long-term dataset of Sumatra’s three most charismatic mammal species distributions. Our dataset, based on 8866 occurrence records of Sumatran elephant, 670 records of Sumatran orangutan, and 1199 records of Sumatran tiger, updates distribution of the three species in Indonesia and enhances the IUCN assessment (*E.m. sumatranus* [5]; *P. abelii* [6]; and *P.t. sumatrae* [7]). However, there are some limitations in this study, where the dataset we use is presence-only, and presence records come from various sources that may not be standardized. However, the relatively large datasets of all habitat types across Sumatra overcome potential sampling bias in this study. Other limitations are related to the covariates used. The environmental variable layers that are important for each species, especially ecological factors such as food plants for herbivores and prey species for carnivores, which are the basic needs of the species, were not included in the analysis due to the scarcity of data.

### 4.1. Model Performance and Utility of Models to Identify Area of Suitable Habitats

The selection of reliable and robust models for projections of species distribution is essential for species conservation and management, as well as for spatial planning [65]. None of the tested models consistently outperformed the others and provided superior predictions across all performance criteria for all three species, a finding in line with other comparative studies of models [66,67,68]. Predictive performance is similar for all discriminant matrices. However, different emphasis on models and the model’s relationship to environmental variables for each species may cause the predicted distribution to vary [67,69].

In our study, for example, the Random Forest and the Ensemble model had similar predictive performance but would have selected very different areas for conserving the three charismatic mammals. Our ensemble of five techniques successfully predicted areas of importance for the three species, including several that had been identified in a different important area project for each charismatic species [70]. The present study showed a comprehensive framework for model assessment regarding fitting performances, species response curves, predictive capacity, and model stability. Here, we used AUC, Kappa, TSS, Jaccard, and Sørensen to evaluate the performance of five species range prediction models (BRT, MXD, RDF, SVM, and ENS) to predict *E.m. sumatranus*, *P. abelii*, and *P.t. sumatrae* ranges. The results showed that RDF and ENS were the top-performing models, with ENS being the best, while BRT and MXD were the low-performing, with the SVM being the worst. As an ensemble machine learning model, RDF can handle data with non-linear relationships, have a high degree of correlation, multidimensional data, and data with missing values. In addition, the RDF model can avoid a reduction in accuracy due to missing data and noise in the training sample when predicting the relationship between the predictor variables and the response variables. In contrast to the RDF model, our results show that the SVM model tends to be insensitive to missing data, and the algorithm is relatively simple. The RDF and ENS models are complex species distribution models and show better predictive performance in processing complex high-dimensional data, such as the data used in this study. However, the results of our study showed that for the three species, no method consistently outperformed others. Our results highlighted the shortcomings and advantages of the models. In particular, ENS was the most reliable method with robust predictions. Furthermore, predictive performances varied more between species than between modeling methods, consistent with previous studies [71,72], which suggests that the individual characteristics of a species should be emphasized when choosing appropriate methods. Based on our results, we recommend the use of multiple modeling approaches to generate more robust predictions for wildlife management [73,74].

### 4.2. Effect of Environmental Variables on the Three Charismatic Species Occurrence

Our findings offer the first spatially explicit model for Sumatra’s three charismatic mammal species. The most important predictor variables for these populations were the land’s protected status, land use type, and climatic factors. Due to human interference, such as poaching and habitat degradation, all three populations have significantly declined over the past few decades [42,75,76,77,78,79], as have their food resources [80]. In Indonesia, massive economic development of business and development sectors such as forestry, livestock farming, agriculture, mining, and settlement development has led to changes in land cover, without exception, on the island of Sumatra. Land conversion for forest concessions and plantations, and settlements are the main cause of deforestation. This, in turn, has exacerbated human–wildlife conflict, hunting, and an overall decrease the quality of life of the three charismatic species, thus causing an increase the death rate of those species beyond their birth rates. It is clear that human interference influenced the population size, and with our models, this drive can shape the distribution of the three mammal species. Substantial declines in all three populations have been noted in various studies. For *E.m. sumatranus*, from a total of 23 elephant habitats in Indonesia, the number of elephants is estimated to be between 928–1379 individuals based on the data compilation of the Indonesian Elephant Conservation Forum (FKGI) in 2019. The number of Sumatran elephants continues to decline from the population estimate in the 1980s (2800–4800 individuals [81,82,83]), to 2007 when it ranged from 2400–2800 individuals [84], and in 2014 when it decreased to 1724 individuals (data compilation by FKGI in 2014). For *P. abelii*, on the basis of new transect surveys in 2015, the population size of *P. abelii* was estimated at 14,613 individuals. To date, the total *P. abelii* population is estimated at 6600 individuals. Current scenarios for future forest loss predict that as many as 4500 individuals could vanish by 2030 [20]. In 2019, *P.t. sumatrae* was estimated to number around 550 adults, which is higher than the population reported in the 2007–2017 Indonesian Tiger Conservation Strategy and Action Plan and other sources, such as the IUCN red list. However, the study conducted during the *P.t. sumatrae* PVA process in 2019 used more complete and up-to-date data than the previous analysis and indicated that the *P.t. sumatrae* population was still experiencing a declining trend. This is based on the fact that many tiger habitats have experienced narrowing, fragmentation, and disturbance. Deforestation on Sumatra is massive (e.g., in central Sumatra: Riau and Jambi Provinces). In 2000, the forest ecosystem area was 7.8 million ha, and in 2014 only 4.4 million ha of forest remained, a decrease of approximately 3.4 million ha (43%). Riau Province is one of the provinces on Sumatra with the highest rate of forest loss and the most extensive; about 4.4 million hectares of the 6.9 million hectares of forest cover (~63%) were lost between 1985 and 2009. Nearly half of Sumatra’s total forest loss between 2000 and 2009 occurred in Riau [42]. Moreover, modeling by [85] predicts that from 2002 to 2016, 34.55% of the forest in central Riau was lost, and from 2016 to 2050, a predicted 58.19% of Sumatran forests will further be lost. Of this area, 82% is in lowland (non-peat) ecosystems, which are the most important habitat for *E.m. sumatranus* and *P.t. sumatrae* [42].

Most of Sumatra’s protected areas are located in isolated highlands and other inhospitable areas for farming [86]. For example, the two largest national parks on the island of Sumatra, Gunung Leuser and Kerinci Seblat, protect almost all of the forests in the remote and inaccessible mountains. The low and even non-existent rate of change in protected areas, when compared to other lands (e.g., logging concessions), contributed to providing a more stable long-term habitat for these three mammal species in Sumatra. Deforestation and land use change mainly occur outside protected areas. Our study found a more suitable area for the three charismatic mammal species in Sumatra than the previous study, but the distribution is narrower than previously reported. The history of creating protected areas in Indonesia is likely the cause of this disparity.

Indonesia’s unique climate is an important part of the ecological picture and illustrates how it affects the distribution of each species in space and time. Climate impacts on species distribution consist of: (i) dynamics in a total suitable area, (ii) optimal environmental changes, and/or (iii) exposure to extinction [87]. In our model, climatic conditions significantly contribute to the distribution of *P. abelii*. Generally, primate’s responses to climate change vary among different species. *P. abelii* and many other arboreal species are adapted to forests that fruit in response to Indonesian weather patterns that are part of global climate systems. Annual and inter-annual cycles of wet and dry seasons are essential for creating the food supply for orangutans and many other forest-dependent frugivores. Climate change could seriously affect *P. abelii* and forests by changing the timing or abundance of fruiting and facilitating fire and flood that destroys habitat. Although the response of tropical forests to climate change is extremely uncertain [88], it is possible that the increased temperatures could lead to considerable forest die-back and an increase in the frequency and severity of forest fires [89,90]. Next, it needs to be noted that the magnitude of climate change effects will be higher in the montane areas or high-altitude ecosystems, such as Leuser Ecosystems, and make these areas more threatened, particularly for conserving *P. abelii* [91,92,93]. Maintaining protected areas means saving wildlife, maintaining a stable temperature and rainfall, and reducing carbon emissions, which trigger global climate change [94].

### 4.3. Contribution of Different Land-Use Types to Three Charismatic Species Conservation in Sumatra

The pace of development that converts the three charismatic mammal’s habitats which we focus on into infrastructure, large-scale plantation and agricultural land, settlements, and so on, on the one hand, benefits the community by increasing welfare and improving the level of the economy, but on the other hand, it also impacts those three mammals and other endangered species. Competition for land and resources is one problem that arises in line with development activities. Habitat narrowing and forest landscape dissolution are risky for the survival of the three mammal species, and this is further exacerbated by human–wildlife conflict incidents, which are increasingly prevalent [95,96,97].

It is essential to understand the different types of land use with the most relevant landscape characteristics to maintain existing populations or increase population sizes of endangered large mammal species. The existence of protected areas has effectively and successfully prevented government-sanctioned industrial-driven massive deforestation [86]. Within the boundaries of protected areas, forest conversion is minimal to other types of land use, such as industrial forest plantations, oil palm, or wood fiber. Concession permits that alter natural forest ecosystems are generally not issued by the Indonesian government within protected area boundaries. Indonesian government policy stipulates that companies do not receive concession rights within protected areas. However, the definition of what is meant by ‘forest’ is often played by some companies to obtain leases on production forests or change forests slightly outside the concession boundaries allocated to them due to blurred boundaries or incoherence of land use maps. The three charismatic mammal species we report here need to be preserved by identifying suitable landscapes next to protected areas.

The rate of deforestation caused by industry in protected forests is minimal. However, land tenure claims to land grabbing around protected areas by agro-businesses often occur and lead to the isolation and destruction of important forest corridors around protected areas (e.g., industrial-driven deforestation around Bukit Tiga Puluh National Park or in the areas known as the Bukit Tiga Puluh Landscape). The impact of land grabbing may push communities or small farmers to the edge of the forest and into protected areas. In addition, almost all industrial operating permits, timber and oil palm concessions are granted in lowland forest, which is mostly the habitat of three charismatic mammals. Between 1990–2000, there was severe forest loss (>1% year^−1^) in 40 PAs in Sumatra dedicated to preserving germplasm (>35% of existing PAs), while 60% had been violated by logging trails, which showed massive forest degradation.

For *E.m. sumatranus* populations, our study showed that 25.87%, 2.97%, 3.37%, and 3.10% of the suitable landscape in Sumatra was mapped in FPC, LC, IOPP, and SFC, respectively. The population of *P. abelii* and *P.t. sumatrae* for FPC, LC, IOPP, and SFC, respectively, were 3.43%, 2.54^−3^%, 0.81%, 2.48%, and 4.31%, 3.94%, 0.48%, 2.24%. There is less than 16% of forest in forest concession areas, and although a relatively small proportion compared to forests in PAs, existing forest concessions have an essential role in preserving and providing refuge areas for the remaining populations of the three species. According to available data, forest concessions may be crucial for maintaining wild populations in tropical regions [98,99]. On the other hand, the existence of forest concessions provides benefits by effectively reducing disturbance and encroachment by smallholders [100]. Well-managed forest concessions can also act as corridors facilitating movement, dispersal, and exchange between viable and non-viable populations. A better understanding of conservation managers needs to be encouraged, especially regarding the synergy and collaboration of stakeholders in managing forest concessions, which also have a very important role in promoting economic development and protecting forest habitats in forest concession areas [100]. However, compared to protected areas, land use changes, including deforestation, are most likely to occur in forest concessions, so stakeholder commitment is needed and must involve proper management, especially for conserving these three species.

### 4.4. Habitat Conservation and Management Recommendations

The distribution map we produce helps provide conclusions about the ecology and conservation of the three focal charismatic species and is also an essential tool for cross-sectoral conservation managers in Sumatra. However, our data may suggest a careful assessment of their status on Sumatra. With this insight into their ecology, we can begin to elucidate further threats they may face and direct conservation measures. Currently, *E.m. sumatranus*, *P. abelii*, and *P.t. sumatrae* utilize habitat at the very edge of the forest and therefore are at greater risk of conflict with humans and experience a higher likelihood of habitat disturbance. Previous research indicates that these species may adjust well to human presence, using the forest as a refuge and exploiting agricultural landscapes for prey. We do not dispute this assertion, but the low presence that we recorded, even at the forest edge, implies that the species is not occurring in high numbers inside of protected areas, and thus, it might not be optimally benefiting from the protection provided by protected areas. Our findings revealed that more than 50% of the three species potential occurrences were found outside of the protected areas, and they may occur in large numbers in agricultural or concession landscapes. Indonesian Academic of Science (LIPI) also reported that around 70% of endangered species occurred outside the protected areas [101]. A previous study also showed that less than 20% of biodiversity hotspots occurred within protected areas in Sumatra [102]. However, they may be more vulnerable than previously thought if they are not utilizing protected areas for protection. This is confirmed by reports of high levels of wildlife–human conflict for these species around several protected areas and indications of retribution killings. Enhancing biodiversity habitats through other effective area-based conservation measures (OECMs) is critical to address conservation targets [103]. Based on this, we suggest that it is necessary to develop maps of forestry and non-forestry concessions connected to the landscapes for these species. Furthermore, we encourage the Ministry of Environment and Forestry as a management authority to carry out internal and external strengthening to ensure that the conservation of these species is accommodated in the Regional Spatial Planning (*Rencana Tata Ruang Wilayah*), both at the provincial and district levels. Therefore, developing corridors to maintain connectivity between protected areas and OECMs is vital for conserving the three species.

The three charismatic mammals we report here have interior distributions; however, they still use and harness habitats at a moderate distance from the forest’s edge. The fact that the species’ habitats are accessible to humans and subject to overexploitation, as well as the other elements of habitats, such as vegetation cover and prey, emphasizes the significance of protection within the protected areas. Although the data used are derived from various large-scale and long-term surveys and monitoring, our records indicate a relatively low presence of data on this species in several protected areas. We suggest the need for initiatives to increase the protection of these species based on actual data considerations. Strengthening the protection of their habitats and the corridors that connect their landscapes to function ecologically and obtain full support from all stakeholders are needed. The critical point is to reduce the risk of inbreeding and loss of genetic diversity by strengthening the network of protected and unprotected areas, preserving habitat connection among populations, and/or translocating individuals into small and isolated populations [104]. Providing opportunities for dispersal to other populations through connecting habitats and protecting population resources are crucial conservation measures for wide-ranging large mammals. [105]. For example, three essential areas in Sumatra, namely Tesso Nilo and the area between Rimbang Baling and Bukit Tigapuluh, are landscapes that play a role in maintaining connectivity that allows movement and exchange between populations, especially to preserve at least two species of charismatic mammals (*E.m. sumatranus* and *P.t. sumatrae*) throughout the study area. Reducing the offtake of the three species through hunting, retaliatory killing, and problem animal removal should be pursued while ensuring strategies for linking between populations can be implemented and scaled up.

## 5. Conclusions

This study contributes important ecological information for these three charismatic mammal species and demonstrates an easily replicable method of examining the distribution of large mammals. The information is essential to implementing conservation initiatives for the species and is crucial for effectively managing protected or non-protected areas that are part of essential ecosystem areas that are interconnected. Moreover, this study offers a research and analysis outline that is generic and easily adaptable for application across the species’ range. To successfully and efficiently protect these three charismatic species within fast-changing habitats, collaboration and analysis of current data throughout their ranges are crucial. Furthermore, recognizing areas where the presence of the three species is most likely to occur gives stakeholders an essential tool for conservation. The model generates a distribution map that describes the predicted areas for each species with a low to a high probability of presence [106]. The large-scale data generated from the distributional maps are beneficial to many stakeholders. They may act as a rough tool to help identify areas of conservation priority and, subsequently, direct ranger patrols, anti-poaching efforts, and anti-encroachment operations. In addition, analyzing spatial data on land cover changes for each species metapopulation is necessary. Furthermore, analysis of spatial data through overlays between the habitats of the three species and ongoing land use plans (forest concessions, mining, and oil palm plantations) should be encouraged to obtain an overview of the conflict conditions in land use in the habitat areas of each species.

## Figures and Tables

**Figure 1 animals-12-02722-f001:**
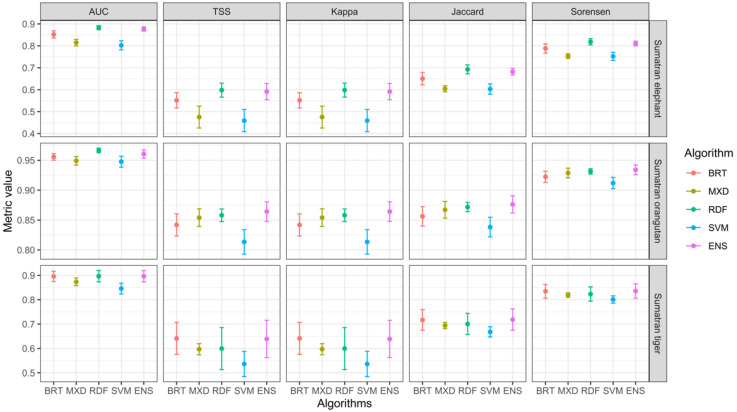
Model evaluation for Sumatran elephant, orangutan, and tiger for various algorithms using discrimination metrics.

**Figure 2 animals-12-02722-f002:**
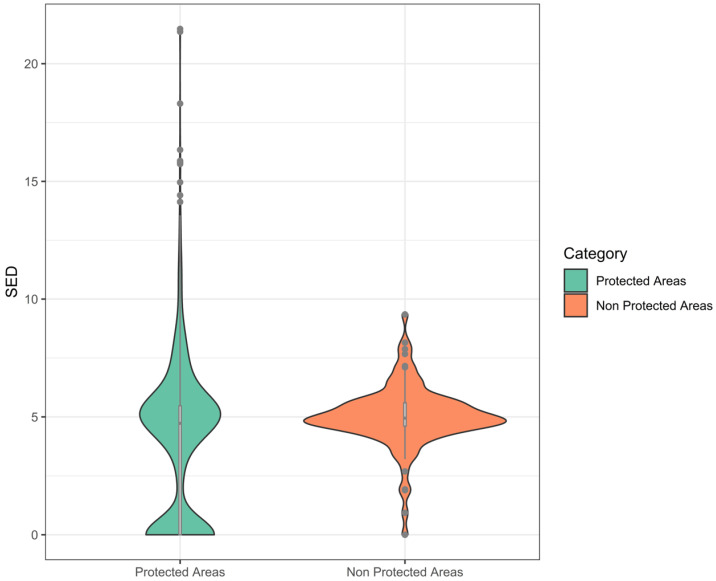
Climatic exposure based on standardized Euclidean distance of climate parameters within protected and non-protected areas.

**Figure 3 animals-12-02722-f003:**
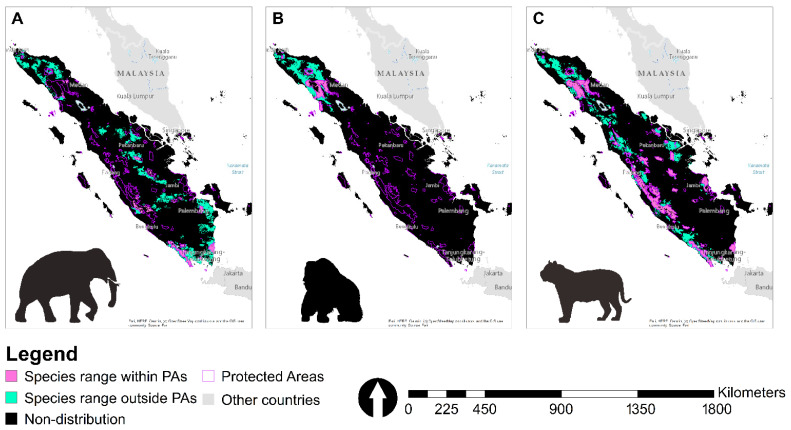
Predictive maps of areas suitable for native populations of *E.m. sumatranus* (**A**), *P. abelii* (**B**), *P.t. sumatrae* (**C**).

**Table 1 animals-12-02722-t001:** Contribution of each variable for the best model (Ensemble).

Group	Variables	Species		
		*E.m. sumatranus*	*P. abelii*	*P.t. sumatrae*
Topographic	Slope	3%	9%	6%
Topographic	Eastness	3%	1%	3%
Topographic	Northness	3%	1%	2%
Biophysical	NDWI	4%	9%	9%
Biophysical	SAVI	2%	1%	2%
Biophysical	Protected areas	14%	4%	15%
Anthropogenic	Land cover	1%	11%	6%
Anthropogenic	Access to FPC	8%	13%	14%
Anthropogenic	Access to LC	21%	5%	8%
Anthropogenic	Access to IOPP	8%	10%	8%
Anthropogenic	Access to SFC	7%	6%	4%
Climatic	Mean temperature	6%	11%	8%
Climatic	Temperature seasonality	7%	14%	6%
Climatic	Annual precipitation	6%	2%	4%
Climatic	Precipitation seasonality	6%	4%	5%

## Data Availability

Not applicable.

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
