# Peer review of "Drivers of Three Most Charismatic Mammalian Species Distribution across a Multiple-Use Tropical Forest Landscape of Sumatra, Indonesia"

_animals, 2022, doi:10.3390/ani12192722_

Round 1
Reviewer 1 Report
I have read with great interest the manuscript of Rahman et al. and I found it to be of considerable validity both from the methodological point of view and for the important implications in the conservation of the three threatened species. The multimodel approach is certainly very useful for identifying the areas of potential presence of species starting from presence-only data and at the same time it is very robust. The manuscript, as far as I can judge, is written in good English but there is some minor typo in the text. I have only three suggestions for the authors:
1) Reading the manuscript it seems that the authors tend to identify the areas predicted as suitable by the models as areas of presence or distribution. It needs to be made clear that suitable areas are only areas of potential presence but that the actual presence is not verified.
2) It would be necessary to have some more details on the statistical analyzes and on the models used.
3) The discussion is too extensive and, although interesting suggestions are given for future management for the conservation of the three species, it should be more closely linked to the results. I suggest reducing it by a quarter of its current length.
Author Response
Response to Reviewer 1 Comments
We would like to thank the reviewer for through reading of this manuscript and for the thougtful comments and constructive suggestions, which help to improve the quality of this manuscript. Our response follows.
Point 1: I have read with great interest the manuscript of Rahman et al. and I found it to be of considerable validity both from the methodological point of view and for the important implications in the conservation of the three threatened species. The multimodel approach is certainly very useful for identifying the areas of potential presence of species starting from presence-only data and at the same time it is very robust. The manuscript, as far as I can judge, is written in good English but there is some minor typo in the text. I have only three suggestions for the authors:
Response 1: We are thankful for the empowered, positive comments to improve our manuscript. We appreciate the time and effort that you have dedicated to providing your valuable feedback on my manuscript. We have already corrected some typo in the maintext and rephrasing several sentences as well.
Point 2: 1) Reading the manuscript it seems that the authors tend to identify the areas predicted as suitable by the models as areas of presence or distribution. It needs to be made clear that suitable areas are only areas of potential presence but that the actual presence is not verified.
Response 2: Thank you for pointing this out. We have already tried our best to collect some reliable data from observations, field survey, and secondary portal data to retrieve species presence data and we have already conducted our analysis based on species distribution modeling protocol, following Zurell et al. (2020) (https://doi.org/10.1111/ecog.04960).
Point 3: 2) It would be necessary to have some more details on the statistical analyzes and on the models used.
Response 3: Thank you for your suggestion. We have already improve our statistical analysis for the species distribution model from the pre-processing (i.e., variable selection), removing autocorrelation from the occurrence data, performing algorithms, and evaluation model as the standard protocol for species distribution model. Then, we also have already used some statistical test to compare our parameters used in this model along with its significance.
Point 4: 3) The discussion is too extensive and, although interesting suggestions are given for future management for the conservation of the three species, it should be more closely linked to the results. I suggest reducing it by a quarter of its current length.
Response 4: Thank you for your suggestion. We have already remove some unnecessary sentences in the discussion section.

Reviewer 2 Report
Thank you for giving me the opportunity to review the manuscript “Drivers of Three Most Charismatics Mammalian Species Distribution across a Multiple-Use Tropical Forest Landscapes of Sumatra, Indonesia”. The manuscript explores the effect of various categories of factors on the distribution of three species of large mammals, aiming to predict their potential range and provide recommendations for species conservation planning and management options. The manuscript brings new information on the distribution and its driving factors of three charismatic and endangered large mammal species of Sumatra being, therefore, of high conservacy interest.
Overall, the manuscript is easy to follow and concise, but there are some grammar issues (some of them identified below), so I would recommend the revision of English language.
Besides, I feel that the emphasis should be placed on the results with ecological and conservation implications, not on the technical aspects concerning the performance of various models. Therefore, I would suggest restructuring the results and discussion parts, bringing forward the potential distribution of the three species and the effects of the considered drivers, revealed by the best models, and adressing afterwards the technical issues and the justification of choosing the ENS model.
One of the main results of the study, valid for all three species, is their lower occurrence probability in protected areas, which is rather counterintuitive. I think this should be discussed in more detail in the text.
I also have a series of minor comments:
The title has two errors: it is ”charismatic mammalian species”, not ”charismatics mammalian species” and you should opt for singular or plural in ”a multiple-use tropical forest landscapes” – delete ”a” or the final ”s” in ”landscapes”.
Line 14: if you use the word ”extinction” it means that the species are completely gone, so there is no point in talking about their potential distribution. You may use instead ”local extinctions”.
Line 17 and elsewhere: „biophysical parameters” instead of „biophysics parameters”.
Line 31: a model cannot improve the distribution, it can enhance the knowledge on the species distribution.
Line 32: “for third populations” replace with “for the distribution of the three species”.
Line 50: “hrabors”.
Line 51: Sirenia sounds strange in this enumeration, for which I actually do not see the point. Dugongs may sometimes enter rivers, but I would not place them here. But maybe I miss something…
Line 53: “there are three highly charismatic and endemic species” – first, the Sumatran elephant and tiger are endemic subspecies, not species, second, these are not the only charismatic endemic endangered large mammals – there is also the Sumatran rhino for example... I would rephrase lines 50-53.
Line 66: “the ecology and population” – the population is one of the main objects of interest for ecology – the population ecology. Rephrase.
Line 70: you can not “assess the distribution and habitat use“ by “by prioritizing critical areas to be protected”. You may “assess the distribution and habitat use“ to prioritize “critical areas to be protected”.
Line 74: you mean “presence-only data”? The presence data has long been important for mapping.
Line 80: “is crucial”.
Line 84: delete “the” before “certain”.
Line 88 and elsewhere: not the study uses algorithms, but the authors in the study.
Line 88: “charismatic”.
Line 92: you should formulate other hypotheses. The first one is rather trivial. These are forest species, therefore it is obvious that they are highly depedent on forests versus other types of land cover.
Line 96: you missed a . before The.
Line 104: the percentage would be more informative than the raw value of the surface.
Line 124: “reports of monitoring projects”.
Line 130: stratified what?
Line 133-134: incomplete sentence.
Line 148: what was the threshold for the correlation coefficient?
Line 164: delete “the” before “most”.
Line 166: maximized Soerensen metric?
Line 192: Jaccard
Line 194: ENS is not presented in the methods section, and the acronym appears for the first time here, without explanation.
Figure 1 is not really relevant for the general reader. Rather Figure 1S.
Line 206: important
Line 208: “predicting” instead of “depicting”.
Line 210: “to occur” instead of “occurred”.
Line 217: you should explain “climatic exposure” in the methods section – what does it mean, how is it expressed, which are its biological and conservation implications.
Lines 221-221: “highest… than” do not match together. Highest is superlative, than asks for a comaparative. Rephrase.
Figure 2 caption: “Climatic exposure based on standardized Euclidean distance” needs to be explained in the methods section.
Line 228: , before “Concessions”
Lines 231-232: “in the lowland areas to montane areas… (Fig. 3A…)” you can not tell from the figure which is lowland.
Line 242: delete “While,”.
Figure 3: increase image resolution
Line 262: “ecological factors” - the term is very vague, give examples; climatic variables are also ecological factors...
Lines 277-279: the sentence is redundant with the first oe in the paragraph. Delete or combine the two.
Line 308: “and the decrease in food quality” - this does not fit in the sentence structure. Rephrase.
Line 310: “livestock” and “settlement” are not economic sectors, but “livestock farming” and “settlement development”.
Line 316-317: “this drive shaped the distribution of the three mammal species” – this is hard to read. Rephrase.
Line 318: “out of” is not the best word choice here.
Line 324: delete “the” before P. abelli.
Line 328: delete “Along with two other charismatic mammals,” – it is confusing.
Lines 328, 333, 473: “sumatrae”
Line 328: “was” instead of “is”.
Lines 335-336, 350: add , before “and” and after “e.g.”
Lines 352-353: “a more suitable area“ - more than what? Rephrase.
Lines 371-372: “particularly for conserving effort the P. abelii” is not understandable – rephrase.
Lines 479-480: “the distribution of large mammals” instead of “large mammalian distribution”.
Figure S1: Why are the values of occurrence likelihood ranging between 0 and 1 for the numerical variables and higher than 1 for the two categorical factors?
Author Response
Response to Reviewer 2 Comments
We would like to thank the reviewer for through reading of this manuscript and for the thougtful comments and constructive suggestions, which help to improve the quality of this manuscript. Our response follows.
Point 1: Thank you for giving me the opportunity to review the manuscript “Drivers of Three Most Charismatics Mammalian Species Distribution across a Multiple-Use Tropical Forest Landscapes of Sumatra, Indonesia”. The manuscript explores the effect of various categories of factors on the distribution of three species of large mammals, aiming to predict their potential range and provide recommendations for species conservation planning and management options. The manuscript brings new information on the distribution and its driving factors of three charismatic and endangered large mammal species of Sumatra being, therefore, of high conservacy interest.
Response 1: Thank you for your positive comments and the suggestion. We appreciate the time and effort that you have dedicated to providing your valuable feedback on my manuscript.
Point 2: Overall, the manuscript is easy to follow and concise, but there are some grammar issues (some of them identified below), so I would recommend the revision of English language.
Besides, I feel that the emphasis should be placed on the results with ecological and conservation implications, not on the technical aspects concerning the performance of various models. Therefore, I would suggest restructuring the results and discussion parts, bringing forward the potential distribution of the three species and the effects of the considered drivers, revealed by the best models, and adressing afterwards the technical issues and the justification of choosing the ENS model.
One of the main results of the study, valid for all three species, is their lower occurrence probability in protected areas, which is rather counterintuitive. I think this should be discussed in more detail in the text.
Response 2: Thank you for thoroughly reading this manuscript and the thoughtful comments and constructive suggestions, which helped improve the manuscript. We have improved the writing style, including a check by independent, native British English speakers with a scientific background. We reinforced the introduction description, methods, discussion, and interpretation by addressing the line-to-line comments below and reviewing the document. For details, see the reply to reviewers 1 and 2. Moreover, we have restructurized our result section as suggested. Regarding the proportion of occurrence within the protected areas, we have already discussed further in the discussion section.
Point 3: The title has two errors: it is ”charismatic mammalian species”, not ”charismatics mammalian species” and you should opt for singular or plural in ”a multiple-use tropical forest landscapes” – delete ”a” or the final ”s” in ”landscapes”.
Response 3: Thank you to point this out. We have already addressed the correction and update the title with “Drivers of Three Most Charismatic Mammalian Species Distribution across a Multiple-Use Tropical Forest Landscape of Sumatra, Indonesia” [Lines 2, 3].
Point 4: Line 14: if you use the word ”extinction” it means that the species are completely gone, so there is no point in talking about their potential distribution. You may use instead ”local extinctions”.
Response 4: Thank you for the suggestion. We continue to use the word "extinction" because of the various threats we mention trigger not only local extinctions but also total extinctions. This is indicated by the loss of the population, particularly for the three species in several main habitats, which can lead to the extinction of all three species in the entire landscape of Sumatra.
Point 5: Line 17 and elsewhere: „biophysical parameters” instead of „biophysics parameters”.
Response 5: The correction has been made [Lines 17, 29, 145, 227].
Point 6: Line 31: a model cannot improve the distribution; it can enhance the knowledge on the species distribution.
Response 6: The correction has been made [Line 33].
Point 7: Line 32: “for third populations” replace with “for the distribution of the three species”.
Response 7: The correction has been made [Line 34].
Point 8: Line 50: “hrabors”.
Response 8: The correction has been made [Line 54].
Point 9: Line 51: Sirenia sounds strange in this enumeration, for which I actually do not see the point. Dugongs may sometimes enter rivers, but I would not place them here. But maybe I miss something.
Response 9: In this sentence, we explain that in Sumatra, there are 13 orders of mammals, these include land mammals and aquatic mammals. The only aquatic mammal is Order Sirenia (Family: Dugongidae, Genus: Dugong, Species: Dugong Dugong dugon).
Point 10: Line 53: “there are three highly charismatic and endemic species” – first, the Sumatran elephant and tiger are endemic subspecies, not species, second, these are not the only charismatic endemic endangered large mammals – there is also the Sumatran rhino for example. I would rephrase lines 50-53.
Response 10: The correction has been made [Lines 56-57].
Point 11: Line 66: “the ecology and population” – the population is one of the main objects of interest for ecology – the population ecology. Rephrase.
Response 11: The sentences have been corrected [Lines 69-72].
Point 12: Line 70: you can not “assess the distribution and habitat use“ by “by prioritizing critical areas to be protected”. You may “assess the distribution and habitat use “to prioritize “critical areas to be protected”.
Response 12: The correction has been made [Lines 73-74].
Point 13: Line 74: you mean “presence-only data”? The presence data has long been important for mapping.
Response 13: The correction has been made [Line 79].
Point 14: Line 80: “is crucial”.
Response 14: The correction has been made [Line 85].
Point 15: Line 84: delete “the” before “certain”.
Response 15: Done [Line 89].
Point 16: Line 88 and elsewhere: not the study uses algorithms, but the authors in the study.
Response 16: The correction has been made [Line 93].
Point 17: Line 88: “charismatic”.
Response 17: The correction has been made [Line 93].
Point 18: Line 92: you should formulate other hypotheses. The first one is rather trivial. These are forest species; therefore, it is obvious that they are highly depedent on forests versus other types of land cover.
Response 18: Thank you for pointing this out. We would like to address how forest ecosystem can strongly influence the species fitness due to in the first hypothesis – thus, deforestation within the areas become crucial for these species existence. Meanwhile, we have already rephrase our first hypothesis [Lines 97-99].
Point 19: Line 96: you missed . before The.
Response 19: Done [Line 102].
Point 20: Line 104: the percentage would be more informative than the raw value of the surface
Response 20: The information has been added (Line 109].
Point 21: Line 124: “reports of monitoring projects”.
Response 21: Done [Line 130].
Point 22: Line 130: stratified what?
Response 22: The correction has been made. We mean we create pseudo-absence data with randomly constrained based on environmental and geographical dataset following Lobo et al. (2010). https://doi.org/10.1111/j.1600-0587.2009.06039.x.
Point 23: Line 133-134: incomplete sentence.
Response 23: The correction has been made. We consider to rephrase the sentences [Lines 139-141].
Point 24: Line 148: what was the threshold for the correlation coefficient?
Response 24: We used 0.7 for the threshold, following Dormann et al. (2013) study in Ecography (https://doi.org/10.1111/j.1600-0587.2012.07348.x) [Line 155]
Point 25: Line 164: delete “the” before “most”.
Response 25: Done [Line 171].
Point 26: Line 166: maximized Soerensen metric?
Response 26: The correction has been made [Line 173].
Point 27: Line 192: Jaccard
Response 27: Done [Line 205].
Point 28: Line 194: ENS is not presented in the methods section, and the acronym appears for the first time here, without explanation.
Response 28: The ENS model has been mentioned in Method Section [Line 164].
Point 29: Figure 1 is not really relevant for the general reader. Rather Figure S1.
Response 29: We thought that its crucial to visualize the model evaluation based on discrimination metrics to the reader in the main text.
Point 30: Line 206: important
Response 30: Done [Line 218].
Point 31: Line 208: “predicting” instead of “depicting”.
Response 31: Done [Line 220].
Point 32: Line 210: “to occur” instead of “occurred”.
Response 32: Done [Line 222].
Point 33: Line 217: you should explain “climatic exposure” in the methods section – what does it mean, how is it expressed, which are its biological and conservation implications.
Response 33: The correction has been made [Lines 192-196].
Point 34: Lines 221-221: “highest… than” do not match together. Highest is superlative, than asks for a comaparative. Rephrase.
Response 34: The correction has been made [Line 234].
Point 35: Figure 2 caption: “Climatic exposure based on standardized Euclidean distance” needs to be explained in the methods section.
Response 35: The method has been updated with the climatic exposure calculation.
Point 36: Line 228: , before “Concessions”.
Response 36: Done [Line 240].
Point 37: Lines 231-232: “in the lowland areas to montane areas… (Fig. 3A…)” you can not tell from the figure which is lowland.
Response 37: The correction has been made [Line 243].
Point 38: Line 242: delete “While,”.
Response 38: Done [Line 254].
Point 39: Figure 3: increase image resolution.
Response 39: The figure has been updated following the image resolution in the Animals Journal writing guidelines. However, transferring the original image to Microsoft Word reduced the image quality.
Point 40: Line 262: “ecological factors” - the term is very vague, give examples; climatic variables are also ecological factors.
Response 40: The examples have been added (Line 275].
Point 41: Lines 277-279: the sentence is redundant with the first one in the paragraph. Delete or combine the two.
Response 41: One of the sentences has been deleted and retains the other [Lines 280-281].
Point 42: Line 308: “and the decrease in food quality” - this does not fit in the sentence structure. Rephrase.
Response 42: The correction has been made [Lines 323-324].
Point 43: Line 310: “livestock” and “settlement” are not economic sectors, but “livestock farming” and “settlement development”.
Response 43: Done [Line 326]
Point 44: Line 316-317: “this drive shaped the distribution of the three mammal species” – this is hard to read. Rephrase.
Response 44: The correction has been made [Lines 332-333].
Point 45: Line 318: “out of” is not the best word choice here.
Response 45: The correction has been made [Line 335].
Point 46: Line 324: delete “the” before P. abelli.
Response 46: Done [Line 341].
Point 47: Line 328: delete “Along with two other charismatic mammals,” – it is confusing.
Response 47: Done [Line 344].
Point 48: Lines 328, 333, 473: “sumatrae”
Response 48: The correction has been made [Lines 345, 349, 501].
Point 49: Line 328: “was” instead of “is”.
Response 49: The correction has been made [Line 345].
Point 50: Lines 335-336, 350: add, before “and” and after “e.g.”
Response 50: Done.
Point 51: Lines 352-353: “a more suitable area“- more than what? Rephrase.
Response 51: The correction has been made [Line 372].
Point 52: Lines 371-372: “particularly for conserving effort the P. abelii” is not understandable – rephrase.
Response 52: The correction has been made [Line 391].
Point 53: Lines 479-480: “the distribution of large mammals” instead of “large mammalian distribution”.
Response 53: The correction has been made [Lines 507-508].
Point 54: Figure S1: Why are the values of occurrence likelihood ranging between 0 and 1 for the numerical variables and higher than 1 for the two categorical factors?
Response 54: Categorical factors represent the frequency of the species occurrence for each categorical value; therefore, the value can be more than 1. On the other hand, continuous variables represent binary probability for the occurrence likelihood for each parameter.
